# Sex Difference in Running Stability Analyzed Based on a Whole-Body Movement: A Pilot Study

**DOI:** 10.3390/sports10090138

**Published:** 2022-09-16

**Authors:** Arunee Promsri

**Affiliations:** 1Department of Physical Therapy, School of Allied Health Sciences, University of Phayao, Phayao 56000, Thailand; arunee.pr@up.ac.th; Tel.: +66-54-466-666 (ext. 3817); 2Unit of Excellence in Neuromechanics, School of Allied Health Sciences, University of Phayao, Phayao 56000, Thailand

**Keywords:** variability, neuromuscular control, movement synergy, sex-specific manner, female, treadmill running, recreational runner, sports-related injury, principal component analysis (PCA), largest Lyapunov exponent

## Abstract

A sex-specific manner in running tasks is considered a potential internal injury risk factor in runners. The current study aimed to investigate the sex differences in running stability in recreational runners during self-preferred speed treadmill running by focusing on a whole-body movement. To this end, principal component analysis (PCA) was applied to kinematic marker data of 22 runners (25.7 ± 3.3 yrs.; 12 females) for decomposing the whole-body movements of all participants into a set of principal movements (PMs), representing different movement synergies forming together to achieve the task goal. Then, the sex effects were tested on three types of PCA-based variables computed for individual PMs: the largest Lyapunov exponent (LyE) as a measure of running variability; the relative standard deviation (rSTD) as a measure of movement structures; and the root mean square (RMS) as a measure of the magnitude of neuromuscular control. The results show that the sex effects are observed in the specific PMs. Specifically, female runners have lower stability (greater LyE) in the mid-stance-phase movements (PM_4__−__5_) and greater contribution and control (greater rSTD and RMS) in the swing-phase movement (PM_1_) than male runners. Knowledge of an inherent sex difference in running stability may benefit sports-related injury prevention and rehabilitation.

## 1. Introduction

Running is one of the most popular sports, but is also associated with a high incidence of musculoskeletal injuries [1,2]. Running injuries are thought to be caused by extrinsic (e.g., shoes and running surface) and intrinsic (e.g., anatomical difference, running experience, performance, and sex-specific manner) factors [2,3]. Interestingly, the prevalence of several types of running injuries (e.g., hamstring injuries, patellofemoral pain, and iliotibial band syndrome) differs between the sexes [2,4]. When considering the sex differences, different performance between males and females (i.e., sex gap) is relatively high in recreational runners [5,6], of which biological differences between the sexes (e.g., hormonal factors, skeletal muscle mass, and oxidative capacities) are accepted as the primary cause [7,8]. Moreover, the sex-specific manner in which individuals perform running tasks is reported [9,10,11] and listed as one potential risk factor for musculoskeletal injuries [2,3]. For instance, a more adducted hip in the stance phase found more in females than in males is considered one possible cause of the high incidence of lower-limb injuries in female runners [12,13]. Hence, analyzing the running biomechanics may beneficially provide an understanding of the underlying injury mechanisms, specifically in terms of the neuromuscular control involved in running movement patterns [14,15,16].

Since the cooperative contribution of whole-body segments is necessary for effectively completing any given motor task [16], determining whole-body movement during running may provide a better understanding of an inherent sex-specific manner in performing running tasks. When focusing on motor behaviors, motor activities have been revealed as a combination of different movement synergies that cooperatively achieve the given task goal [17]. These task-dependent movement synergies can be flexibly adapted to internal and external demands [16]. Dimensionality reduction methods (e.g., principal component analysis, PCA) applied to kinematic marker data gained popularity in decomposing movement synergies from whole-body movements [17,18]. This method can help with redundancy issues of motor apparatus by reducing the number of features needed to complete the task, by creating fewer new variables while still maintaining most of the information about how people move from the original feature set [17,18]. Specifically, PCA applied to kinematic data yields a set of one-dimensional movement components or synergies called “principal movements” (PM) [17,19]. Information about individual PMs’ position and acceleration reflects their direct link to forces in the system and myoelectric activity [15,16], providing an effective assessment of neuromuscular control of individual movement strategies [14,15,16,17].

The application of non-linear methods, e.g., local dynamic stability quantified by the determination of the largest Lyapunov exponent (LyE), are applied to analyze the ability to compensate for small internal or external perturbations to maintain functional locomotion [20,21,22,23]. The term “stability” is commonly defined as the ability of a system (e.g., the movement system) to maintain its original state in the face of internal (e.g., neuromuscular) and external (e.g., environmental) disturbances [22], in which stability parameters provide information regarding the noise in the motor task performance and explicitly quantify the performance of the dynamic error correction [24]. Conversely, the term “variability” is often used and defined as an indirect measure of how stable a person performs the motor tasks (e.g., running), since the noise in the motor task or in the environment can move a person’s dynamic state closer to their stability limits [24,25]. Hence, in terms of local dynamic stability, running stability can be referred to as the ability of the neuromuscular system to control infinitesimal perturbations during locomotion [20,21,22,23]. Furthermore, decreased running stability is reported to be associated with a reduced ability to compensate for small perturbations that may raise the risk of overuse injuries, e.g., bone stress injuries caused by repetitive monotonous loads that exceed bone loading capacity [26,27]. In this sense, applying the LyE to individual PMs’ positions can help to quantify the stability of individual movement strategies forming together to complete the running tasks [25].

In summary, the current study aimed to investigate running stability in terms of the difference in neuromuscular control of the main movement synergies of treadmill running between male and female recreational runners. PCA was used to characterize movement synergies (i.e., principal movements, PMs) from whole-body movements. PCA-based variables were then determined to differentiate sex differences in running stability. As previously reported [9,10,11], males and females have distinct kinematic running characteristics. Therefore, it was hypothesized that the sex difference in gait stability would manifest in the specific PMs relevant to the current task.

## 2. Materials and Methods

### 2.1. Secondary Data Analysis

The kinematic marker data of 22 healthy recreational runners with treadmill running experiences used in the current study were obtained from a peer-reviewed open-access dataset [28]. In order to calculate the sample size, the original data article used a *priori power analysis* based on ten previous studies assessing the sex differences in running movement patterns [10,29,30,31,32,33,34,35,36,37], yielding an average effect size (Cohen’s d) for kinematic comparisons between males and females of 1.25 [9]. Based on this computation, with a significance level of α = 0.05 and a desired power = 0.8, the suggested sample size was N = 24 [9]. However, in the open-access data port [28], only the datasets of 23 participants were provided, and the data from one participant had a problem with the incomplete recording, leading to only 22 participants’ data being recruited for analysis in the current study. The characteristics of the participants are shown in Table 1. All participants had self-reported no lower extremity injuries that required medical consultation and/or sports participation disruption for longer than two weeks in the last six months prior to study participation [9]. Furthermore, the Board for Ethical Questions in Science of the University of Innsbruck, Austria, approved the study protocol in accordance with the ethical principles of the Helsinki Declaration (Certificate No. 70/2019), and all participants provided written informed consent before participation [9].

Measurement procedures were fully described in Mohr et al. [9]. In brief, thirty-nine reflective markers were distributed over all body segments of the volunteers according to the standard marker setup “Plug-In Gait” (Vicon Motion Systems Ltd., Oxford, UK), and a safety harness was attached to the volunteer’s waist to avoid the risk of injury in the case of a fall or slip before measurement. First, each volunteer was instructed to complete a 10 min warm-up period (5 min brisk walking and 5 min tested speed running). Then, three-dimensional (3D) marker trajectories for 30 s running with a sampling rate of 250 Hz were captured by an 8-camera motion tracking system (Vicon Bonita B10 with Nexus 2.9.2 software: Vicon Motion Systems Ltd., Oxford, UK).

### 2.2. Movement Synergy Extraction

All data processing was conducted in MATLAB version 2022a (MathWorks Inc., Natick, MA, USA). For each dataset, nine asymmetrical markers placed on the upper arms, lower arms, right scapular, upper thighs, and lower thighs were omitted, and the remaining 30 markers contributing 90 spatial coordinates (x, y, z) were interpreted as 90 dimensional posture vectors. Each dataset was pre-processed centered by subtracting the mean posture vector [17], normalized to the mean Euclidean distance [17], and weighted by considering sex-specific mass distributions [38]. The data from all volunteers were concatenated to form a 165,000 × 90 input matrix (250 [sampling rate] * 1 [number of trials] * 30 [trial duration] * 22 [number of participants] x 90 [marker coordinates]) for further PCA. An example of the original running movements derived from one male participant is represented in Appendix A.

PCA was calculated using a singular-value decomposition of the covariance matrix through the PManalyzer software [17] to decompose all kinematic data into a set of orthogonal eigenvectors (i.e., principal components: PC; *k* denotes the order of movement component). Animated stick figures can be created to characterize each eigenvector’s movement pattern, which has been called “principal movement” (PM_k_) [17]. Moreover, the actual time evolution (time series) of individual PM_k_ is quantified by the PC scores or “principal positions” (PP_k_(*t*)), since they represent positions in posture space, i.e., the vector space spanned by the PC-eigenvectors [17]. The word ‘‘principal” in the variable names denotes that these variables were obtained through a PCA, and (*t*) indicates that these variables are functions of time *t* [17]. In regards to Newton’s mechanics, the second-time derivatives, “PM_k_-accelerations or principal accelerations” (PA_k_(*t*)), can be calculated from the PP_k_(*t*) according to the conventional differentiation rules [17]. The associations between PA_k_(*t*) and leg myoelectric activity were demonstrated for postural control tasks [15], supporting that PA-based variables could reveal the neuromuscular control of each PM_k_ [16,39,40,41]. Before PCA-based variable computation, PM-time series were filtered with a 3rd-order zero-phase 5-Hz-low-pass Butterworth filter, to avoid noise amplification in the differentiation processes [41], and were normalized to the individual’s running speed [42].

The current study used leave-one-out cross-validation to evaluate the vulnerability of the PM_k_ and the dependent variables to changes in the input data matrix to address validity considerations [17]. The first five PCs proved robust, explained 97.3% of the total variance, and were selected to test the hypotheses.

### 2.3. PCA-Based Variable Computation

In order to determine the sex effects in the running variability, the subject-specific *largest Lyapunov exponent* (LyE) of PP_k_(*t*) was used. This is a non-linear analysis method computed by quantifying the rate of divergence of close trajectories in state space, i.e., the ability of the motor system to attenuate small perturbations revealed as the divergence of the trajectories in state space (Figure 1) [25,43]. LyE was calculated by applying Wolf’s algorithm [44], of which the time delay (*τ* = 10) and embedding dimension (*m* = 4) were determined using the average mutual information (AMI) [25,43] and the false nearest neighbor algorithms [45], respectively. A greater LyE value reflects an inability of the motor system to diminish the perturbations [21], resulting in a higher divergence of the state space trajectories that reflects the lower individual’s running stability.

Moreover, the current study also investigated the sex differences in the movement structures or compositions [17] and the magnitude of neuromuscular control [42] of individual PMs. In this sense, two more PCA-based variables were selected. First, the subject-specific *relative standard deviation* (rSTD_k_) of PP_k_(*t*) was calculated as a measure of the relative contribution of PM_k_ to the deviation movement of the trial (i.e., a measure of the total amount of postural movement) by computing the percentage of contribution of individual PP_k_(*t*) STD to the total amount of postural movement [17]. Second, the subject-specific *root mean square* (RMS) of PA_k_(*t*) was calculated as a measure of the magnitude of neuromuscular control [42].

### 2.4. Statistical Analysis

The IBM SPSS Statistics software version 26.0 (SPSS Inc., Chicago, IL, USA) was used for all statistical analyses, with the alpha level set at *a* = 0.05. Shapiro–Wilk tests suggested an independent sample t-test for testing sex differences in three PCA-based variables (PP_k__rSTD, PP_k__LyE, and PA_k__RMS) as measures of running stability. Cohen’s *d* was computed as the effect size.

## 3. Results

### 3.1. Movement Synergies

Table 2 represents the eigenvalues of individual PMs representing the absolute contribution of PM_k_ to overall variance and the descriptive movement characteristics of the first five PMs (PM_1–5_), whose visualizations and animated stick figures are represented in Figure 1 and Appendix A, respectively.

The first PM resembles the swing-phase movement. The second and third PMs are associated with both leg movements, whereas the fourth and fifth PMs are mainly related to the mid-stance phase movements.

### 3.2. Sex Differences in Running Stability

As shown in Table 3, the main findings show that the sex differences in running stability assessed through three PCA-based variables (PP_k__LyE, PP_k__rSTD, and PA_k__RMS) are observed only in the specific PM. Specifically, an analysis of running variability (PP_k__LyE) shows that females have higher variability in the mid-stance-phase movement components (PP_4__LyE [PM_4_; *p* = 0.015] and PP_5__LyE [PM_5_; *p* = 0.047]) than males.

Moreover, an analysis of movement structure or composition (PP_k__rSTD) and an analysis of the magnitude of neuromuscular control (PA_k__RMS) shows that females have a greater contribution (PP_1__rSTD [*p* = 0.005]) and a higher neuromuscular control (PA_1__RMS [*p* = 0.016]) in the swing-phase movement (PM_1_) than males.

## 4. Discussion

The current study investigated the running stability of male and female recreational runners during preferred speed treadmill running by focusing on the whole-body movement, which was decomposed into a set of movement synergies (i.e., principal movements, PMs) using principal component analysis (PCA). Three PCA-based variables—PP_LyE, PP_rSTD, and PA_RMS—were computed for each PM as measures of the movement structures, variability, and magnitude of neuromuscular control, respectively. As expected, the main findings show that the sex effects on running stability appear in the specific PMs. Specifically, females have higher variabilities of the single-limb-support phase movement components (PM_4_ and PM_5_). In addition, females have a higher contribution to the swing-phase movement component (PM_1_), with a greater magnitude of neuromuscular control of this movement component, than males.

Running is considered analogous to bouncing, since when the foot strikes the ground, kinetic and gravitational potential energy are momentarily stored as elastic strain energy components (e.g., muscles, tendons, and ligaments), which are virtually recovered during the propulsive second part of the stance phase [46]. In other words, mechanical energy is absorbed each step by muscle–tendon units when the body decelerates during the brake, and is restored when the body reaccelerates during the push-off [47]. In this sense, the higher variability of the mid-stance phase movement components (PM_4_ and PM_5_) found in female runners than in male runners may be related to the ability to control the weight-bearing caused by lower-limb muscle strength. Specifically, it is reported that physically active young females have lower hamstring torque than males, while no sex difference is observed in the quadriceps torque [14,48]. During running, the hamstrings have three main functions related to the weight-bearing and take-off phases of the running cycle [4]: (1) decelerating the knee extension at the end of the forward swing phase of the gait cycle through an eccentric contraction providing dynamic stabilization to the weight-bearing knee; (2) facilitating hip extension through an eccentric contraction at foot strike to stabilize the weight-bearing leg; and (3) supporting the gastrocnemius in extending the knee during the take-off phase of the running cycle. These interpretations agreed with the previous notions that hamstring injuries occur during the stance phase [49,50].

When considering the swing-phase movement component (PM_1_), female runners also have greater neuromuscular control and a higher contribution to this movement component than male runners. These findings could demonstrate that the sex differences might be influenced by muscle activity, since PMs’ acceleration is associated with lower-limb myoelectric activity [15]. However, since movement synergies were extracted from whole-body movements [17,41], running tasks require coordination of all body segments. In this movement synergy, both the lower limbs and the core and the upper limbs that cooperatively work together may involve greater neuromuscular control [16].

In summary, the main empirical findings of the current study suggest that the sex differences in running stability should be considered for training, injury prevention, and rehabilitation. Mainly, the high instability of the sub-phase “mid-stance” of the gait cycle (PM_4_ and PM_5_) in female runners might indicate different neuromuscular control influenced by different muscular strength between the sexes [51], especially in the hamstring muscles [14,48]. Moreover, the whole-body movement analysis reveals the cooperative work of all the body segments to effectively achieve the task goal, demonstrating that the neuromuscular system controls posture through muscles that produce relative movements between body segments [15,52]. In this sense, not only the neuromuscular control training of the lower extremities, but also the other parts of the body (e.g., the core and upper extremities as mainly seen in PM_1_), should be considered for reducing the sex gap in running stability, since the differences in the strength of major muscle groups of the body between the sexes are reported in healthy individuals [51].

### Limitations and Future Study

Although one limitation of the current study is the relatively small sample size (10 females and 12 males), the findings may be useful as a pilot study. Another limitation is that the physical activity participation and running habits of the participants could not be statistically tested for sex differences, since the original data were recorded on the ratio scale [28]. Hence, only the analysis of descriptive statistics was performed. However, both the female and male groups were physically active. Adding or considering these points in future research may explain the impact of physical activity and running habits on running performance.

Typically, changes in gait parameters while running are associated with an increasing intensity of muscle activation [53], of which, lower-limb muscle strength has been marked as one potential risk factor for sports-related injuries, with females being the most affected [54]. Moreover, sex differences in muscle recruitment patterns and synergies are well-reported, and speculated to contribute to the increased injury risk in female athletes. In this sense, a combination study of muscle activity and self-specific manners related to running stability is of interest for future research.

## 5. Conclusions

The current study highlights the effectiveness of decomposing the whole-body movement into movement components/strategies, i.e., principal movements (PMs), in determining the sex differences in running stability. Specifically, female runners have higher variability in the mid-stance phase (PM_4_ and PM_5_) movement components, and more contribution and control of the swing-phase (PM_1_) movement component than male runners, implying that the inherent impacts of sex differences in running stability should be considered for sports-related injury prevention and rehabilitation.

## Figures and Tables

**Figure 1 sports-10-00138-f001:**
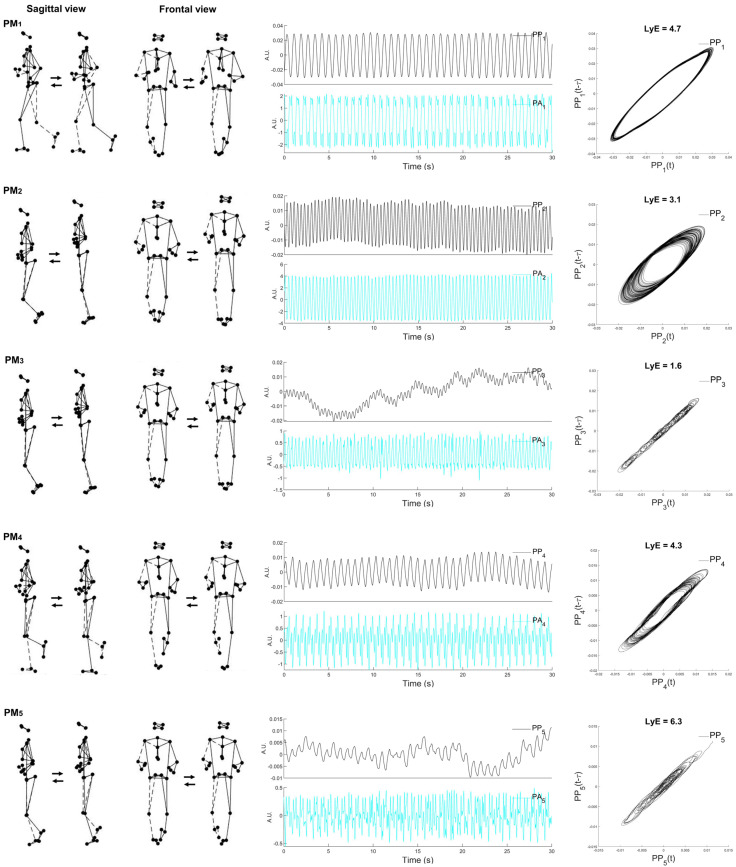
Visualization of the first five principal movements (PM_1–5_; (**left column**)), examples of principal position (PP_k_) and principal acceleration (PA_k_) (**middle column**) over time, and the space–time representation for the calculated Lyapunov exponent (LyE) of PP_1__−__5_ (**right column**). Note: the dashed line represents the right limb. Movements are more precise and can be easily characterized when viewed in an animated stick figure video (Appendix A). The example data are derived from one male participant.

**Table 1 sports-10-00138-t001:** Characteristics of participants (M = mean ± SD; * *p* < 0.001).

	Total	Male (n = 10)	Female (n = 12)
Age (yrs.)	25.7 ± 3.3	26.8 ± 2.8	24.6 ± 3.5
Weight (kg)	67.9 ± 9.3	74.7 ± 6.8	61.2 ± 5.2 *
Height (cm)	174.7 ± 8.5	180.3 ± 4.8	169.2 ± 7.1 *
Body mass index (kg/m^2^)	22.3 ± 2.1	23.0 ± 2.2	21.4 ± 2.0
Preferred running speed (km/h)	10.4 ± 1.0	10.8 ± 1.1	10.1 ± 0.8
Physical activity (h/week)	(%)	(%)	(%)
10–20	34.8	60.0	15.4
5–10	52.2	30.0	69.2
1–5	13.0	10.0	15.4
Running habits (h/week)	(%)	(%)	(%)
5–10	13.6	0	25
1–5	72.7	80	66.7
0	13.6	20	8.3

**Table 2 sports-10-00138-t002:** The eigenvalue (%) and descriptive movements of the first five principal movements (PM_1–5_) analyzed from all participants when performing self-preferred speed treadmill walking.

PM*_k_*	Eigenvalues (%)	Descriptive Movements
1	60.7	The swing phase: anti-phase arm and leg movements in the sagittal plane combined with trunk rotation
2	16.1	Both hip and knee flexion and extension movements combined with whole-body movements in the vertical direction
3	11.3	Both knee flexion and extension combined with the anteroposterior sliding of the treadmill
4	5.2	The mid-stance phase: anti-phase arm and leg movements
5	4.0	The mid-stance phase: anti-phase arm and leg movements combined with the mediolateral sliding of the treadmill

**Table 3 sports-10-00138-t003:** Comparisons of the largest Lyapunov exponent (PP_k__LyE), relative standard deviation of PP (PP_k__rSTD), and root mean square (PA_k__RMS) of the first five principal movements (PM_1–5_) between male and female runners (mean ± SD; * *p* < 0.05; *p*-values smaller than 0.05 are printed in bold).

PP_k__LyE	Male	Female	*p*-Value	Effect Size	Observed Power
1	5.1 ± 0.8	5.5 ± 0.9	0.312	0.470	0.613
2	3.8 ± 0.9	3.8 ± 0.5	0.913	0.000	0.500
3	2.2 ± 0.8	2.3 ± 0.4	0.642	0.158	0.514
4	3.5 ± 0.6	4.2 ± 0.7	**0.015 ***	**1.074**	**0.845**
5	5.7 ± 2.0	7.5 ± 2.0	**0.047 ***	**0.900**	**0.790**
**PP_k__rSTD**	**Male**	**Female**	** *p* ** **-Value**	**Effect Size**	**Observed Power**
1	26.9 ± 1.9	29.4 ± 1.9	**0.005 ***	**1.315**	**0.903**
2	14.4 ± 1.3	14.6 ± 0.8	0.652	0.185	0.519
3	12.4 ± 3.4	11.4 ± 2.7	0.461	0.326	0.558
4	8.5 ± 0.6	8.0 ± 0.8	0.102	0.707	0.715
5	7.6 ± 1.3	6.6 ± 1.1	0.053	0.830	0.764
**PA_k__RMS**	**Male**	**Female**	** *p* ** **-Value**	**Effect Size**	**Observed Power**
1	1.7 ± 0.2	1.9 ± 0.1	**0.016 ***	**1.265**	**0.893**
2	3.2 ± 0.2	3.3 ± 0.2	0.085	0.500	0.625
3	0.6 ± 0.1	0.6 ± 0.1	0.919	0.000	0.500
4	0.6 ± 0.1	0.6 ± 0.1	0.337	0.000	0.500
5	0.3 ± 0.1	0.4 ± 0.1	0.115	1.000	0.824

## Data Availability

Not applicable.

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
