# Peer review of "Sex Difference in Running Stability Analyzed Based on a Whole-Body Movement: A Pilot Study"

_sports, 2022, doi:10.3390/sports10090138_

Round 1
Reviewer 1 Report
First, I would like to congratulate the excellent work that was presented. I understand how laborious it is to prepare a manuscript for publication and we must give proper credit. Congratulations.
You've done a good job. However, I have a few points to raise.
Introduction
The paper addresses the sex difference in running stability. However, in the introduction there is little information about sex differences, running stability, and the methods that will be used. Please I suggest adding such information.
Lines 33-37
Please, you mentioned, "...performing several athletic tasks between male and female have been listed as one internal risk factor for musculoskeletal sports injuries...". Please could you mention what these tasks are and why it is an internal risk factor? Please provide more reasons. Also, you mentioned, "...understand the underlying injury mechanisms...". What mechanisms?
Line 79
Please, can you add the sample size calculation and how it was done?
Line 195
Please, I would suggest adding a sentence with the main findings of the study.
Discussion
As I mentioned earlier, in the introduction there was little talk about sex differences. In the discussion, there is also a lack of information about it. Therefore, I suggest, please, add more information about the main findings and sex differences.
I suggest you do an analysis of the amount of running hours per week among the participants. I was curious as to whether race time per week will influence such responses. You have participants with 0 h/week, 1-5 h/week, and 5-10 h/week. Please let me know if this point is relevant to you and tell me your reasons.
Limitation
I would suggest changing the title: Sex difference in running stability analyzed based on a whole-2 body movement: a pilot study
Author Response
Response to Reviewer 1 Comments
Point 1: First, I would like to congratulate the excellent work that was presented. I understand how laborious it is to prepare a manuscript for publication and we must give proper credit. Congratulations. You've done a good job. However, I have a few points to raise.
Response 1:
Thank you very much for your valuable time reviewing the manuscript and providing valuable suggestions. I gratefully appreciate all your suggestions and applied them to improve the manuscript. Please find the point-by-point of my responses in the following and in the revised manuscript.
Point 2: Introduction
The paper addresses the sex difference in running stability. However, in the introduction there is little information about sex differences, running stability, and the methods that will be used. Please I suggest adding such information.
Response 2:
Thank you for pointing out how to improve the manuscript. In the revised manuscript, I added more information about the sex differences by mainly rewriting this part located in the first paragraph of the manuscript. In addition, I improved the second and third parts of the Introduction by explaining more about the Method used in the study and summarizing the meaning of running stability that is often described in previous studies based on the local dynamic stability.
Point 3: Lines 33-37
Please, you mentioned, "...performing several athletic tasks between male and female have been listed as one internal risk factor for musculoskeletal sports injuries...". Please could you mention what these tasks are and why it is an internal risk factor? Please provide more reasons. Also, you mentioned, "...understand the underlying injury mechanisms...". What mechanisms?
Response 3:
Regarding these suggestions, I rewrote the first paragraph of the Introduction by adding more information about sex differences in the revised manuscript in both general differences in performance and in a sex-specific manner in performing running. Moreover, I rewrote a bit of the sentence about the injury mechanisms by trying to explain one possible mechanism of the injuries “neuromuscular control”. This issue can be indirectly assessed by investigating motor behavior. This point is linked to the method used in the current study, which is represented in the second paragraph of the Introduction. Briefly, the principal component analysis applied to the kinematic marker data provides a set of different movement strategies called Principal movement (PM), revealing several movement patterns forming together to achieve the given task goal. PCA also yields kinematic information about the position and acceleration of individual PMs. This information has been proved that they directly linked to the forces and myoelectric activities. Hence, PCA-based variables can be used to determine the neuromuscular control of individual movement components/strategies (i.e., PMs).
Point 4: Line 79
Please, can you add the sample size calculation and how it was done?
Response 4:
In the revised manuscript, I added sentences to explain how the original data article calculated the sample size. Briefly, they calculated the sample size data using a priori power analysis based on previous research studies investigating the running movement patterns.
Point 5: Line 195
Please, I would suggest adding a sentence with the main findings of the study.
Response 5:
Thank you very much. I added the suggested sentence in the revised manuscript accordingly.
Point 6: Discussion
As I mentioned earlier, in the introduction there was little talk about sex differences. In the discussion, there is also a lack of information about it. Therefore, I suggest, please, add more information about the main findings and sex differences.
I suggest you do an analysis of the amount of running hours per week among the participants. I was curious as to whether race time per week will influence such responses. You have participants with 0 h/week, 1-5 h/week, and 5-10 h/week. Please let me know if this point is relevant to you and tell me your reasons.
Response 6:
Thank you for pointing out how to improve the manuscript. In the revised manuscript, I added one more paragraph to summarize the main findings and discuss the sex differences observed in the current study by adding one more paragraph in the Discussion.
Regarding the information on running habits, I cannot apply the statistical analysis (e.g., independent sample t-test) to analyze the differences between sexes since the original data were reported on the ratio scale. Therefore, in the revised manuscript, I added this information in the part of the Limitation.
Point 7: Limitation
I would suggest changing the title: Sex difference in running stability analyzed based on a whole-2 body movement: a pilot study
Response 7:
Thank you very much for suggesting improving the title. I agreed with your idea and changed the title of the manuscript accordingly.
Reviewer 2 Report
This study aims to assess the sex difference in running stability analyzed based on whole body movement.
The study is quite interesting and relevant and one of the rare studies I don’t have any comments or suggestions to give to improve it further.
The author did an excellent job,
Author Response
Response to Reviewer 2 Comments
Point 1: This study aims to assess the sex difference in running stability analyzed based on whole body movement.
The study is quite interesting and relevant and one of the rare studies I don’t have any comments or suggestions to give to improve it further.
The author did an excellent job,
Response 1: Thank you very much for your valuable time reviewing the manuscript.
Round 2
Reviewer 1 Report
Very good improvements were done! I really appreciate your effort to improve the article's quality.
Congratulations!